**Data Availability Statement:** The linked SEER-Medicare data used for this analysis are not public use data files. In light of the sensitive nature of the

# Predictors of chemoradiotherapy versus single modality therapy and overall survival among patients with unresectable, stage III non-small cell lung cancer

**Priyanka Bobbili**[1]*, **Kellie Ryan**[2], **Maral DerSarkissian**[1], **Akanksha Dua**[1], **Christopher Yee**[1], **Mei Sheng Duh**[1], **Jorge E. Gomez**[3]

1 Analysis Group, Inc., Boston, Massachusetts, United States of America, 2 AstraZeneca, Gaithersburg, Maryland, United States of America, 3 Icahn School of Medicine at Mt. Sinai, New York, New York, United States of America

* Bobbili@analysisgroup.com

## Abstract

### Introduction

Concurrent chemoradiotherapy (cCRT) was the standard of care for patients with unresectable stage III non-small cell lung cancer (NSCLC) prior to the PACIFIC trial, however, patients also received single modality therapy. This study identified predictors of therapy and differences in overall survival (OS).

### Methods

This retrospective study included stage III NSCLC patients aged ≥65 years, with ≥1 claim for systemic therapy (ST) or radiotherapy (RT) within 90 days of diagnosis, identified in SEER-Medicare data (2009–2014). Patients who had overlapping claims for chemotherapy and RT ≤90 days from start of therapy were classified as having received cCRT. Patients who received sequential CRT or surgical resection of tumor were excluded. Predictors of cCRT were analyzed using logistic regression. OS was compared between therapies using adjusted Cox proportional hazards models.

### Results

Of 3,799 patients identified, 21.7% received ST; 26.3% received RT; and 52.0% received cCRT. cCRT patients tended to be younger (p <0.001), White (p = 0.002), and have a good predicted performance status (p<0.001). Patients who saw all three specialist types (medical oncologist, radiation oncologist, and surgeon) had increased odds of receiving cCRT (p<0.001). ST and RT patients had higher mortality risk versus cCRT patients (hazard ratio [95% CI]: ST: 1.38 [1.26–1.51]; RT: 1.75 [1.61, 1.91]); p<0.001).

data, maintaining patient and provider confidentiality is a primary concern of the National Cancer Institute, SEER, and Centers for Medicare and Medicaid Services. Investigators are required to obtain approval for specific research questions in order to obtain the data. Data are available from the National Cancer Institute (https:// healthcaredelivery.cancer.gov/seermedicare/ obtain/) for researchers who meet the criteria for access to these data. To submit requests for SEER-Medicare data, the following website can be used, and the application can be sent to Elaine Yanisko (yaniskoe@imsweb.com). URL: https:// healthcaredelivery.cancer.gov/seermedicare/ contact.html.

**Funding:** This research was funded by AstraZeneca, Gaithersburg, MD. As detailed in the submission, Dr. Jorge Gomez is an employee of Mount Sinai Hospital and has received funding from AstraZeneca. Kellie Ryan received support in the form of salaries from AstraZeneca, and is a stockholder of AstraZeneca. Mei Sheng Duh, Akanksha Dua, Maral DerSarkissian, Christopher Yee, and I are employees of Analysis Group, Inc., a consultancy that has received research funding from AstraZeneca for this and other studies. All co-authors had a role in the conception and study design, administrative support, data analysis and interpretation, decision to publish, preparation and approval of the manuscript. AD, CY, PB, MSD, and MD had a role in the collection and assembly of data. The specific roles of these authors are articulated in the 'author contributions' section.

**Competing interests:** As detailed in the submission, Dr. Jorge Gomez is an employee of Mount Sinai Hospital and has received funding from AstraZeneca. Kellie Ryan received support in the form of salaries from AstraZeneca, and is a stockholder of AstraZeneca. Mei Sheng Duh, Akanksha Dua, Maral DerSarkissian, Christopher Yee, and I are employees of Analysis Group, Inc., a consultancy that has received research funding from AstraZeneca for this and other studies. This does not alter our adherence to PLOS ONE policies on sharing data and materials.

## Conclusions

Several factors contributed to treatment selection, including patient age and health status, and whether the patient received multidisciplinary care. Given the survival benefit of receiving cCRT over single-modality therapy, physicians should discuss treatment within a multidisciplinary team, and be encouraged to pursue cCRT for patients with unresectable stage III NSCLC.

## Introduction

Non-small cell lung cancer (NSCLC) accounts for 80%-85% of all newly diagnosed lung cancer cases [1]. NSCLC patients present with stage III disease in approximately one third of all cases, and of those cases, only a minority are considered surgically resectable [2–5]. Patients with stage III NSCLC have a 5-year survival rate ranging from 13%-36% [2, 4–6].

While the recent development of novel therapeutic agents has improved the treatment landscape for advanced NSCLC, limited progress has been made in the treatment of early and locally advanced disease [7]. Treatment options for patients with stage III NSCLC can involve multiple modalities, such as combinations of surgery, radiotherapy, and chemotherapy. For several decades, the standard of care for patients with unresectable stage III NSCLC has been platinum-based doublet chemotherapy administered concurrently with radiotherapy (i.e., concurrent chemoradiotherapy or cCRT) with curative intent, followed by active surveillance [8–11]. cCRT combines the benefits of local regional control achieved by single-modality radiotherapy and the metastatic risk reduction achieved by single-modality chemotherapy to provide improved results in stage III patients [12–14]. A 2010 meta-analysis reported an 8% reduction in risk of death for cCRT compared to single-modality radiotherapy, and a 13% reduction when compared to sequential CRT [14]. Several phase III clinical trials and meta-analyses have further established that cCRT is superior to both sequential CRT and single-modality radiotherapy [13–15].

More recently, results from a randomized, phase III, placebo-controlled trial demonstrated that patients with stage III NSCLC who were treated with durvalumab, a selective, high-affinity, anti-PD-L1 human IgG1 monoclonal antibody, as consolidation therapy following cCRT, experienced significantly better survival outcomes compared to placebo [13]. Durvalumab was approved by the US Food and Drug Administration (FDA) in February 2018 to treat patients with inoperable stage III NSCLC whose cancer has not progressed following treatment with platinum-based cCRT and has become the new standard of care [16].

The administration of specialized multimodality therapy requires integrative knowledge on prognostic and predictive factors associated with survival in this patient population [17]. Ideally, these complex treatment decisions should be made by a multidisciplinary team (MDT) of physicians and care-givers in close coordination, as it has been shown that health outcomes improve for NSCLC patients when treatment is directed by an MDT [17, 18].

Despite current guidelines and research, some patients with stage III NSCLC are treated with single modality chemotherapy or radiotherapy instead of cCRT. Eligible patients who do not receive cCRT may also not receive benefit of newly approved targeted treatments, such as durvalumab, which is indicated for patients who have received CRT. Given the potential clinical benefit of cCRT, and now durvalumab, which is administered after cCRT, it is important to understand why patients with unresectable stage III NSCLC do not receive cCRT.

The objective of the present retrospective analysis using the Surveillance, Epidemiology, and End Results (SEER)-Medicare database was conducted to investigate factors, including receipt of multidisciplinary care, which can impact a patient's receipt of cCRT versus single modality therapy, to identify potential opportunities to improve patient care and to identify differences in survival across treatment groups.

## Materials and methods

### Study design and data source

This retrospective, longitudinal cohort study was conducted using the SEER-Medicare database, which links data from the SEER cancer registries to Medicare enrollment and claims files. The overall methods are similar to a previously published study [19]. SEER-Medicare data was further linked to the American Medical Association (AMA) Physician Masterfile using encrypted national provider index (NPI) data [20]. The study period spanned from January 1, 2009 to December 31, 2014, the latest SEER-Medicare data available at the time of analysis. The index date was defined as the date of initiation of the first therapeutic regimen following a diagnosis of unresected stage III NSCLC. The observation period ranged from the index date until the earliest of end of eligibility, end of data availability (December 31, 2014), or patient death.

### Study population

The study population consisted of patients diagnosed with NSCLC between 2009 and 2013, identified using the 2-digit cancer site recode in SEER (C34.0-C34.9 'Lung and bronchus') and International Classification of Disease for Oncology (ICD-O-3) morphology codes 8012, 8046, 8070, 8071, 8140, 8250, 8480, 8481, 8490, and 8570. Patients were further required to have stage III disease at the time of diagnosis, based on the American Joint Committee on Cancer (AJCC) Tumor, Nodes, Metastases (TNM) staging system, 6th edition (for patients diagnosed in 2009) or 7th edition (for patients diagnosed in 2010–2013). Eligible patients were required to be 65 years or older at the time of diagnosis, have coverage for both Medicare Part A and Part B, and have no enrollment in a health maintenance organization (HMO). Patients were additionally required to have at least six months of continuous insurance eligibility immediately prior to the index date, in order to collect baseline information, and at least one month of continuous insurance eligibility immediately following the index date. Lastly, patients were required to have a claim for any NSCLC treatment within 90 days of their NSCLC diagnosis, identified by Healthcare Common Procedure Coding System (HCPCS) codes or Current Procedural Technology (CPT) codes, or by National Drug Codes (NDC) for oral therapy [21].

Patients who had multiple primary tumors or surgical tumor resection were excluded from the analysis. Surgical resection was identified using (1) SEER registries, if a cancer-directed surgery was performed within 4 months of the date of NSCLC diagnosis, and (2) surgical procedures in Medicare claims. Patients were excluded if the reporting source of their lung cancer diagnosis was an autopsy or death certificate. This study was determined to be exempt from IRB review by the New England Independent Review Board as the research involved analysis of secondary de-identified data.

### Identification of treatment cohorts

Treatment cohorts were classified based on the initial treatment regimens received by patients [22]. Single modality therapy was defined as treatment consisting of either only systemic therapy (ST) or radiotherapy (RT). If the first medical claim following NSCLC diagnosis was for a

chemotherapy or targeted therapy agent, patients were considered to have received ST only if there were no claims for RT in the 90 days following the chemotherapy or targeted therapy claim. If the first medical claim was for RT, patients were considered to have received RT only if there were no claims for chemotherapy or targeted therapy in the 90 days following the RT claim.

Patients were considered to have received CRT if they had claims for both chemotherapy and RT within 90 days of initial NSCLC diagnosis or a claim for chemotherapy within 90 days of initial NSCLC diagnosis and a claim for RT within 45 days after the end of chemotherapy. Patients were required to have at least two consecutive claims for RT to be included in the CRT population. The maximum duration of gap between chemotherapy and RT claims that still constituted CRT, defined as 45 days in this study, was based on prior literature as well as discussions with a thoracic oncologist. Among patients who received CRT, cCRT was defined as an overlap in claims, or a gap of up to 14 days between chemotherapy and RT claims [23, 24]. Patients who received treatment with sequential CRT (sCRT) were excluded from this analysis, due to small sample size.

## Demographic, clinical, and diagnostic variables

Baseline characteristics were evaluated during the 6 months prior to the index date. Patient demographic characteristics collected at diagnosis included age, sex, race, and region. Clinical information included tumor characteristics such as histology, grade, and laterality. Metastatic sites were evaluated using ICD-9 codes for malignant neoplasms (these would be considered metastases since patients with more than 1 primary tumor site, i.e., other than NSCLC, were excluded). Potential metastatic in stage III NSCLC patients include the lung and lymph nodes, as well as the heart, large blood vessels near the heart, the diaphragm, backbone, or trachea. [25] Medicare claims data from the baseline period were used to assess Charlson Comorbidity Index (CCI) and predicted performance status (PS); the latter was estimated using a previously validated multivariate logistic regression model that predicted a "good" PS (defined as Eastern Cooperative Oncology Group [ECOG] score 0–2, or Karnofsky Performance Scale [KPS] score 60–100), or "poor" PS (ECOG score 3–5, or KPS score 0–50) per post-2009 recommendations of the American Society of Clinical Oncology [26]. In addition, data from the AMA Physician Masterfile was used to collect information on self-reported clinical specialty of treating physicians.

## Classification of physician specialties

An "initial physician", defined as the attending or referring physician associated with a Medicare claim for a diagnostic radiological procedure during the baseline period, from within 12 weeks before NSCLC diagnosis to 6 weeks after diagnosis, was identified based on a previously developed algorithm to capture the physician who was initially involved in the management of the patient's NSCLC [27]. Specialist physicians seen after the "initial physician" visit were classified as either (1) medical oncologists, (2) radiation oncologists, or (3) surgical specialists according to the primary specialty assigned to a given NPI in the AMA Physician Masterfile. Each NPI in the dataset was assigned only one primary specialty. Physicians who did not have any of these primary specialties were not classified and were not considered as specialists in this analysis.

## Statistical analyses

Descriptive analyses assessing patient baseline and clinical characteristics were conducted using means, standard deviations (SDs), and medians for continuous variables and frequencies

and proportions for categorical variables. Logistic regression models were used to evaluate baseline demographic and clinical predictors of receiving cCRT compared to single modality therapy (either ST or RT), as well as cCRT compared to ST alone and RT alone. A stepwise backwards elimination technique was used to identify significant baseline demographic and clinical predictors with a p-value cutoff of 0.20 for variable removal. Odds ratios were estimated from the logistic regression model, after adjusting for baseline covariates (the variables age, race, sex, stage 3A vs. 3B, setting of residence, histology, ischemic heart disease, COPD, CKD, and proportion of patients with tumor marker tests were forced in the logistic regression models; other variables that were included for selection, but not forced into the model, were index year, region, tumor site, tumor grade, laterality, type of cancer, specific comorbidities, comorbidity index, and predicted PS). Overall survival (OS), defined as the time from the index date to the date of death, was analyzed using the Kaplan-Meier method. Patients were censored if they did not die during the observation period. Adjusted analysis was conducted to compare OS across treatment cohorts using Cox proportional hazards models. Statistical significance was defined based on alpha = 0.05. All statistical analyses were conducted in SAS Enterprise Guide Version 7.1 (Cary, NC).

## Results

### Study population

Between 2009 and 2014, 5,313 patients met eligibility criteria (Fig 1). Of these 272 (5.1%) patients classified as initiating sCRT and 497 (9.4%) patients not classifiable by the algorithms into a treatment cohort were excluded from further analyses. In addition, claims data for 403 patients (7.6%) could not be linked to the AMA Physician Masterfile data and 342 patients (6.4%) did not have an identifiable initial physician, and thus were also excluded. The final analytical sample consisted of 3,799 patients, of which 823 patients (21.7%) initiated treatment with ST, 1000 (26.3%) initiated treatment with RT, and 1976 (52.0%) initiated treatment with cCRT.

### Baseline patient demographic and clinical characteristics

Across all treatment cohorts, the average age of patients was 74.9 years, 47.4% were female, and 84.1% were White (Table 1). The proportions of patients who initiated treated with either ST or RT appeared to decrease between 2009 and 2013. While a smaller proportion of patients initiated cCRT versus ST or RT in 2009, a larger proportion of patients initiated cCRT as their first therapeutic regimen in 2012 and 2013, though these differences reached statistical significance only in 2012 (2012: 21.6% versus 15.3% and 16.8%, respectively, p<0.001). Compared to the ST and RT cohorts, patients in the cCRT cohort were younger (mean age 73.5 years versus 75.5 years and 77.2 years, respectively, p < 0.001), more likely to be male (45.5% versus 49.6% and 49.4%, respectively, p = 0.045), and more likely to be White (86.0% versus 82.6% and 81.4%, respectively, p = 0.002). The average CCI score for the cCRT cohort was lower than either the ST and RT cohorts (5.3 versus 5.8 and 5.6, respectively; p < 0.001). Similarly, the proportion of patients with a "poor" predicted PS was lower for the cCRT cohort than either the ST and RT cohorts (33.5% versus 41.7% and 57.3%, respectively; p < 0.001). Across treatment cohorts, the most common comorbidities were hypertension (78.4%), COPD (75.6%), dyslipidemia (65.3%), ischemic heart disease (41.6%), and diabetes (33.7%). Patients who initiated ST were more likely to have ≥1 tumor marker test during the baseline period compared to patients who initiated RT or cCRT (cCRT: 6.0%; ST: 10.2%; RT: 3.7%; p<0.001). Among those tested, the most common tests administered during baseline were EGFR, molecular

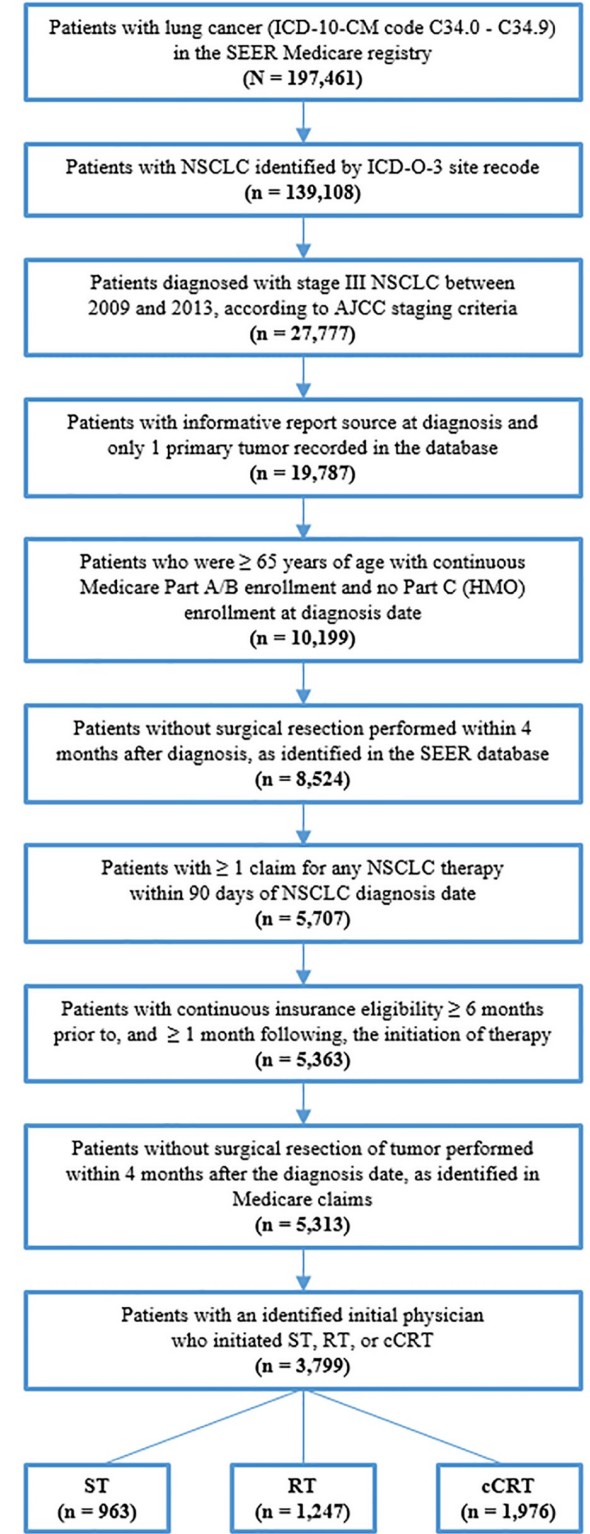

**Fig 1. Sample selection (January 2009–December 2014).**

**Table 1. Comparison of baseline characteristics of unresected, stage III NSCLC patients treated with systemic therapy, radiotherapy, and concurrent chemoradiotherapy[1,2].**

| | All patients (N = 3,799) | Systemic therapy only (N = 823) | Radiotherapy only (N = 1,000) | Concurrent chemoradiotherapy (N = 1,976) | P-value[3] |
|---|---|---|---|---|---|
| **Characteristics assessed at diagnosis** | | | | | |
| **Age at diagnosis, years; mean (SD) [median]** | 74.9 ± 6.4 [74.0] | 75.5 ± 6.7 [75.0] | 77.2 ± 7.0 [77.0] | 73.5 ± 5.4 [73.0] | <0.001* |
| **Female, N (%)** | 1,800 (47.4%) | 408 (49.6%) | 494 (49.4%) | 898 (45.4%) | 0.045* |
| **Race/Ethnicity, N (%)** | | | | | |
| White | 3,194 (84.1%) | 680 (82.6%) | 814 (81.4%) | 1,700 (86.0%) | 0.002* |
| Black | 379 (10.0%) | 74 (9.0%) | 138 (13.8%) | 167 (8.5%) | <0.001* |
| Asian | 134 (3.5%) | 43 (5.2%) | 28 (2.8%) | 63 (3.2%) | 0.010* |
| Other[4] | 92 (2.4%) | 26 (3.2%) | 20 (2.0%) | 46 (2.3%) | 0.257 |
| **Region, N (%)** | | | | | |
| West | 1,236 (32.5%) | 324 (39.4%) | 318 (31.8%) | 594 (30.1%) | <0.001* |
| South | 1,241 (32.7%) | 222 (27.0%) | 354 (35.4%) | 665 (33.7%) | <0.001* |
| Northeast | 818 (21.5%) | 208 (25.3%) | 194 (19.4%) | 416 (21.1%) | 0.008* |
| Midwest | 504 (13.3%) | 69 (8.4%) | 134 (13.4%) | 301 (15.2%) | <0.001* |
| **Setting of residence, N (%)[5]** | | | | | |
| Big metro | 1,910 (50.3%) | 454 (55.2%) | 515 (51.5%) | 941 (47.6%) | <0.001* |
| Metro | 1,174 (30.9%) | 229 (27.8%) | 307 (30.7%) | 638 (32.3%) | 0.066 |
| Urban | 238 (6.3%) | 63 (7.7%) | 51 (5.1%) | 124 (6.3%) | 0.081 |
| Less Urban | 374 (9.8%) | 57 (6.9%) | 102 (10.2%) | 215 (10.9%) | 0.005* |
| Rural | 103 (2.7%) | 20 (2.4%) | 25 (2.5%) | 58 (2.9%) | 0.673 |
| **Primary tumor site, N (%)** | | | | | |
| Main bronchus | 225 (5.9%) | 21 (2.6%) | 68 (6.8%) | 136 (6.9%) | <0.001* |
| Lower lobe | 964 (25.4%) | 248 (30.1%) | 269 (26.9%) | 447 (22.6%) | <0.001* |
| Mid-lobe | 146 (3.8%) | 24 (2.9%) | 34 (3.4%) | 88 (4.5%) | 0.109 |
| Upper lobe | 2,123 (55.9%) | 405 (49.2%) | 553 (55.3%) | 1,165 (59.0%) | <0.001* |
| Other[6] | 341 (9.0%) | 125 (15.2%) | 76 (7.6%) | 140 (7.1%) | <0.001* |
| **AJCC Stage** | | | | | |
| IIIA | 2,260 (59.5%) | 418 (50.8%) | 632 (63.2%) | 1,210 (61.2%) | <0.001* |
| IIIB | 1,539 (40.5%) | 405 (49.2%) | 368 (36.8%) | 766 (38.8%) | <0.001* |
| **Grade** | | | | | |
| Well differentiated | 94 (2.5%) | 31 (3.8%) | 23 (2.3%) | 40 (2.0%) | 0.024* |
| Moderately differentiated | 644 (17.0%) | 130 (15.8%) | 185 (18.5%) | 329 (16.6%) | 0.271 |
| Poorly differentiated | 1,168 (30.7%) | 219 (26.6%) | 307 (30.7%) | 642 (32.5%) | 0.009* |
| Other[7] | 1893 (49.8%) | 443 (53.8%) | 485 (48.5%) | 965 (48.8%) | 0.034* |
| **Histology (IDC-O-3), N (%)[8]** | | | | | |
| Adenocarcinoma (8140) | 1,366 (36.0%) | 408 (49.6%) | 286 (28.6%) | 672 (34.0%) | <0.001* |
| Squamous cell (8070) | 1,608 (42.3%) | 243 (29.5%) | 480 (48.0%) | 885 (44.8%) | <0.001* |
| General NSCLC (8046) | 490 (12.9%) | 90 (10.9%) | 147 (14.7%) | 253 (12.8%) | 0.057 |
| Other | 335 (8.8%) | 82 (10.0%) | 87 (8.7%) | 166 (8.4%) | 0.409 |
| **Characteristics assessed during baseline period** | | | | | |
| **Index year of first therapeutic regimen, N (%)** | | | | | |
| 2009 | 851 (22.4%) | 242 (29.4%) | 231 (23.1%) | 378 (19.1%) | <0.001* |
| 2010 | 740 (19.5%) | 160 (19.4%) | 200 (20.0%) | 380 (19.2%) | 0.882 |
| 2011 | 743 (19.6%) | 153 (18.6%) | 197 (19.7%) | 393 (19.9%) | 0.726 |
| 2012 | 720 (19.0%) | 126 (15.3%) | 168 (16.8%) | 426 (21.6%) | <0.001* |
| 2013 | 698 (18.4%) | 133 (16.2%) | 186 (18.6%) | 379 (19.2%) | 0.167 |

*(Continued)*

**Table 1.** (Continued)

| | All patients (N = 3,799) | Systemic therapy only (N = 823) | Radiotherapy only (N = 1,000) | Concurrent chemoradiotherapy (N = 1,976) | P-value[3] |
|---|---|---|---|---|---|
| **Number of unique metastatic sites,[9] mean (SD) [median]** | 0.8 ± 0.9 [1.0] | 0.9 ± 1.0 [1.0] | 0.7 ± 0.9 [0.0] | 0.9 ± 0.9 [1.0] | <0.001* |
| **Charlson Comorbidity Index,[10] mean (SD) [median]** | 5.5 ± 2.5 [5.0] | 5.8 ± 2.5 [6.0] | 5.6 ± 2.5 [5.0] | 5.3 ± 2.4 [5.0] | <0.001* |
| **Comorbidities present during baseline period, N (%)** | | | | | |
| Hypertension | 2,979 (78.4%) | 659 (80.1%) | 808 (80.8%) | 1,512 (76.5%) | 0.012* |
| COPD | 2,871 (75.6%) | 579 (70.4%) | 806 (80.6%) | 1,486 (75.2%) | <0.001* |
| Dyslipidemia | 2,481 (65.3%) | 551 (67.0%) | 589 (58.9%) | 1,341 (67.9%) | <0.001* |
| Ischemic heart disease | 1,580 (41.6%) | 337 (40.9%) | 457 (45.7%) | 786 (39.8%) | 0.008* |
| Diabetes | 1,279 (33.7%) | 300 (36.5%) | 350 (35.0%) | 629 (31.8%) | 0.036* |
| Cerebrovascular disease | 908 (23.9%) | 193 (23.5%) | 273 (27.3%) | 442 (22.4%) | 0.011* |
| Heart failure | 716 (18.8%) | 174 (21.1%) | 273 (27.3%) | 269 (13.6%) | <0.001* |
| Chronic kidney disease | 452 (11.9%) | 107 (13.0%) | 161 (16.1%) | 184 (9.3%) | <0.001* |
| No comorbidities | 94 (2.5%) | 14 (1.7%) | 22 (2.2%) | 58 (2.9%) | 0.129 |
| **Tumor marker tests during baseline period[11]** | | | | | |
| Patients with ≥1 test, N (%) | 239 (6.3%) | 84 (10.2%) | 37 (3.7%) | 118 (6.0%) | <0.001* |
| **Predicted performance status, N (%)[12]** | | | | | |
| Good | 2,631 (57.9%) | 554 (57.5%) | 530 (42.5%) | 1,547 (66.3%) | <0.001* |
| Poor | 1,913 (42.1%) | 409 (42.5%) | 717 (57.5%) | 787 (33.7%) | <0.001* |

* Significant at the 5% level

**Abbreviations**:

NSCLC: non-small cell lung cancer; SD: standard deviation

[1] The baseline period is defined as the six months prior to initiation of the first therapeutic regimen for NSCLC. Initiation of the first therapeutic regimen was required to occur within 90 days after the initial NSCLC diagnosis.

[2] Patients who were determined to have only received systemic therapy had claims for targeted therapy or chemotherapy within 90 days of their initial NSCLC diagnosis, and no claims for radiotherapy during the same period. Patients who were determined to have only received radiotherapy had radiotherapy claims within 90 days of their initial NSCLC diagnosis, and no claims for targeted therapy or chemotherapy during the same period. Concurrent chemoradiotherapy (cCRT) patients had claims for radiotherapy along with either targeted therapy or chemotherapy, within 90 days of their initial NSCLC diagnosis.

[3] Kruskal-Wallis tests were used to compare categorical variables. Analysis of variance (ANOVA) tests were used to compare continuous variables.

[4] Other category includes Hispanic, Native American, Unknown and races classified as "Other" by SEER-Medicare data.

[5] Setting of residence classifications are based on Rural-Urban Continuum Codes that distinguish metropolitan (metro) counties by the population size of their metro area, and nonmetropolitan counties by degree of urbanization and adjacency to a metro or rural areas.

[6] Other category includes primary tumor site of "Overlap" and "Lung, unspecified".

[7] Other category includes "Undifferentiated" or "Unknown" grade.

[8] Only the three most frequently observed histology types in the study population are reported in this table.

[9] Metastatic sites were evaluated based on ICD-9 codes. Diagnoses of other malignancies were considered to be metastases since all patients in the study population were required to only have 1 primary tumor (NSCLC). Metastatic sites included but were not limited to diagnosis codes for lymphatic and hematopoietic tissue cancer, respiratory cancer (excluding cancer of the lungs), bone and bone marrow cancer, brain and spinal cord cancer, tissue cancer, and endocrine cancer among others.

[10] CCI was calculated based on the method described in Quan et al. (2005). Source: Quan H, Sundararajan V, Halfon P et al. Coding Algorithms for Defining Comorbidities in ICD-9-CM and ICD-10 Administrative Data. Medical Care 2005;43:1130–1139.

[11] Tumor marker tests during baseline period include EGFR gene, KRAS gene, BRAF gene, molecular cytogenetics (e.g., FISH for anaplastic lymphoma kinase or ROS-1 gene arrangement), morphometric analysis, tumor immunohistochemistry (e.g., programmed death-ligand 1, programmed cell death protein 1), multiple gene panel, lung cancer panel, and proteomic testing panel.

[12] Predicted performance status was calculated using age at diagnosis, COPD status, number of inpatient stays, any outpatient visit, number of ED visits, any DME claim, and any prescription drug dispensing during the six month baseline period, based on the method described in Salloum et al. (2011). Good predicted performance status was assigned to patients with ≥ 70% probability of having an Eastern Cooperative Oncology Group (ECOG) score of 0–2 or a Karnofsky Performance Scale (KPS) of 100–60.

**Table 2. Comparison of physician specialists seen by unresected, stage III NSCLC patients prior to initiating treatment with systemic therapy, radiotherapy, and concurrent chemoradiotherapy.**

| | All patients (N = 3,799) | Systemic therapy only (N = 823) | Radiotherapy only (N = 1,000) | Concurrent chemoradiotherapy (N = 1,976) | P-value[3] |
|---|---|---|---|---|---|
| **Any type of specialist seen** | | | | | |
| Medical oncologist | 2,967 (78.1%) | 724 (88.0%) | 703 (70.3%) | 1,540 (77.9%) | <0.001* |
| Radiation oncologist | 1,625 (42.8%) | 123 (14.9%) | 564 (56.4%) | 938 (47.5%) | <0.001* |
| Surgical specialist | 1,816 (47.8%) | 422 (51.3%) | 398 (39.8%) | 996 (50.4%) | <0.001* |
| None of the above | 307 (8.1%) | 51 (6.2%) | 98 (9.8%) | 158 (8.0%) | 0.019* |
| **Only one type of specialist seen**[1] | | | | | |
| Medical oncologist | 869 (22.9%) | 302 (36.7%) | 179 (17.9%) | 388 (19.6%) | <0.001* |
| Surgical specialist | 200 (5.3%) | 36 (4.4%) | 51 (5.1%) | 113 (5.7%) | 0.336 |
| **Two types of specialists seen**[2] | | | | | |
| Medical oncologist and radiation oncologist | 624 (16.4%) | 42 (5.1%) | 236 (23.6%) | 346 (17.5%) | <0.001* |
| Medical oncologist and surgical specialist | 798 (21.0%) | 311 (37.8%) | 108 (10.8%) | 379 (19.2%) | <0.001* |
| **All three types of specialists seen** | | | | | |
| Medical oncologist, radiation oncologist, and surgical specialist | 676 (17.8%) | 69 (8.4%) | 180 (18.0%) | 427 (21.6%) | <0.001* |
| **Number of specialists seen, mean SD [median]** | 1.7 ± 0.9 [2.0] | 1.5 ± 0.7 [2.0] | 1.7 ± 0.9 [2.0] | 1.8 ± 0.9 [2.0] | <0.001* |

\* Significant at the 5% level

**Abbreviations**:

NSCLC: non-small cell lung cancer; SD: standard deviation

[1] <5% of patients saw only a radiation oncologist prior to initiating treatment with systemic therapy, radiotherapy, and concurrent chemoradiotherapy.

[2] <5% of patients saw only a radiation oncologist prior and surgical specialist prior to initiating treatment with systemic therapy, radiotherapy, and concurrent chemoradiotherapy.

cytogenetics (FISH) for ALK or ROS-1, tumor immunochemistry panels (PD-1, PD-L1) and combination lung cancer panels (EGFR, KRAS, ALK, ROS-1, and PD-L1).

Prior to the initiation of treatment, medical oncologists were most frequently seen (78.1%) followed by surgical specialists (47.8%) then radiation oncologists (42.8%) (Table 2). Across cohorts, 676 patients (17.8%) were seen by all three types of physician specialists before the start of treatment. A larger proportion of patients in the cCRT cohort saw all three types of specialists compared to either the ST or RT cohorts (21.6% versus 8.4% and 18.0%, respectively; p < 0.001).

## Predictors of treatment

Several baseline characteristics were associated with receipt of cCRT versus single modality therapy (i.e., either ST or RT) (Table 3). Variables that decreased the odds of receiving cCRT versus single modality therapy included increasing age (OR = 0.93, p < 0.001), being female (OR = 0.85, p = 0.034), being Black (compared to being White; OR = 0.71, p = 0.006), and having a higher CCI score (OR = 0.90, p < 0.001). Conversely, variables that increased the odds of receiving cCRT instead of single modality therapy included a predicted performance status of "good" (OR = 1.72, p < 0.001), and having seen all three types of specialists prior to the initiation of treatment (compared to having only seen one type; OR = 1.87, p < 0.001).

**Table 3. Logistic regression to identify predictors of concurrent chemoradiotherapy versus single-modality therapy among patients with unresected, stage III NSCLC patients.**

| Patient Characteristics | Logistic regression selected variables[1] | |
| --- | --- | --- |
| | Odds Ratio (95% CI) | P-value |
| Age | 0.93 (0.92, 0.94) | <0.001* |
| Female | 0.85 (0.74, 0.99) | 0.034* |
| Race | | |
| White | Ref | |
| Asian | 1.13 (0.74, 1.73) | 0.568 |
| Black | 0.71 (0.55, 0.91) | 0.006* |
| Other | 0.99 (0.61, 1.59) | 0.963 |
| Specialists visited[2] | | |
| One type | Ref | |
| Two types | 1.16 (0.98, 1.36) | 0.081 |
| Three types | 1.87 (1.51, 2.32) | <0.001* |
| Region | | |
| West | Ref | |
| Midwest | 1.59 (1.25, 2.04) | <0.001* |
| Northeast | 1.25 (1.01, 1.54) | 0.038* |
| South | 1.06 (0.86, 1.29) | 0.589 |
| Setting of residence | | |
| Big metro | Ref | |
| Metro | 1.20 (1.01, 1.42) | 0.035* |
| Urban | 1.15 (0.84, 1.58) | 0.370 |
| Less Urban | 1.18 (0.90, 1.54) | 0.244 |
| Rural | 1.16 (0.74, 1.84) | 0.515 |
| Primary tumor site | | |
| Upper lobe | Ref | |
| Lower lobe | 0.72 (0.61, 0.86) | <0.001* |
| Lung, unspecified | 0.63 (0.48, 0.83) | <0.001* |
| Main bronchus | 1.30 (0.94, 1.80) | 0.106 |
| Mid-lobe | 1.42 (0.96, 2.11) | 0.078 |
| Overlap | 0.66 (0.31, 1.42) | 0.285 |
| Grade | | |
| Well differentiated | Ref | |
| Moderately differentiated | 1.31 (0.80, 2.14) | 0.277 |
| Poorly differentiated | 1.63 (1.01, 2.62) | 0.045* |
| Undifferentiated | 2.41 (1.01, 5.76) | 0.048* |
| Unknown | 1.27 (0.79, 2.03) | 0.324 |
| Histology | | |
| Adenocarcinoma | Ref | |
| Squamous cell | 1.28 (1.08, 1.52) | 0.005* |
| General NSCLC | 1.16 (0.92, 1.48) | 0.215 |
| Other | 1.10 (0.83, 1.45) | 0.519 |
| Stage | | |
| 3A | Ref | |
| 3B | 0.83 (0.72, 0.97) | 0.020* |
| Index year of first therapeutic regimen | | |
| 2009 | Ref | |

*(Continued)*

**Table 3.** (Continued)

| Patient Characteristics | Logistic regression selected variables[1] | |
|---|---|---|
| | Odds Ratio (95% CI) | P-value |
| 2010 | 1.13 (0.90, 1.42) | 0.285 |
| 2011 | 1.32 (1.05, 1.65) | 0.019* |
| 2012 | 1.54 (1.22, 1.95) | <0.001* |
| 2013 | 1.43 (1.13, 1.82) | 0.003* |
| 2014 | 0.79 (0.40, 1.56) | 0.498 |
| Metastatic site codes[3] | | |
| Lip, oral cavity, and pharynx | 2.02 (0.78, 5.23) | 0.145 |
| Lymphatic and hematopoietic tissue | 1.91 (1.02, 3.56) | 0.043* |
| Genitourinary organs | 1.71 (1.39, 2.12) | <0.001* |
| Bone and bone marrow | 0.51 (0.34, 0.76) | <0.001* |
| Adrenal gland | 0.42 (0.14, 1.25) | 0.119 |
| Charlson Comorbidity Index | 0.90 (0.86, 0.94) | <0.001* |
| Comorbidities present during baseline period | | |
| Dyslipidemia | 1.37 (1.17, 1.61) | <0.001* |
| COPD | 1.04 (0.87, 1.25) | 0.654 |
| Ischemic heart disease | 1.03 (0.88, 1.21) | 0.693 |
| Chronic kidney disease | 0.87 (0.68, 1.10) | 0.242 |
| Heart failure | 0.78 (0.63, 0.98) | 0.032* |
| Proportion of patients with ≥1 tumor marker tests | 0.86 (0.63, 1.17) | 0.345 |
| Predicted performance status | | |
| Poor | Ref | |
| Good | 1.72 (1.46, 2.02) | <0.001* |

**Abbreviations**: NSCLC: non-small cell lung cancer; CRT: Chemoradiotherapy; CI: Confidence interval

* Significant at the 5% level

[1] A stepwise backwards elimination technique was used to identify significant baseline characteristics with a p-value cutoff of 0.20 for variable removal. Odds ratios were estimated from a logistic regression model adjusting for the following baseline covariates: race, gender, age, region, setting of residence, primary tumor site, grade, laterality, histology, index year of first therapeutic regimen, number of unique metastatic sites, all cancer metastatic sites at baseline, all comorbidities present at baseline, CCI, tumor marker tests, and predicted performance status.

[2] Specialists included were medical oncologists, radiation oncologists, and surgery specialists.

[3] Identified based on the following ICD-9 codes: 140–149, 170, 179–189, 194.0, 196, 198.5, 198.6, 198.7, 198.82, 200–208.

## Overall survival

Patients who received cCRT had a median [IQR] OS of 14.7 months [6.6–34.2 months], whereas ST and RT patients had a median OS of 11.0 months [5.1–22.7 months] and 8.0 months [3.0–17.7 months], respectively (p <0.001) (Fig 2). The 12-month to 60-month mortality rates for cCRT patients were consistently lower compared to patients who received ST or RT. Patients who received ST or RT were at higher risk of mortality compared to patients who received cCRT (HR [95% CI]: ST: 1.38 [1.26–1.51]; RT: 1.75 [1.61, 1.91]); p<0.001) (Table 4). The results remained significant after adjusting for baseline covariates.

## Discussion

This retrospective cohort study assessed differences in baseline characteristics, survival outcomes, and predictive factors for treatment among unresected stage III NSCLC patients aged

| | Total N | Total number of events | Median [IQR] time to death (months) | Mortality rate | | | | |
|---|---|---|---|---|---|---|---|---|
| | | | | 12 months | 24 months | 36 months | 48 months | 60 months |
| Concurrent CRT | 1,976 | 1,399 | 14.7 [6.6, 34.2] | 43.0% | 65.3% | 75.8% | 81.3% | 85.1% |
| Systemic therapy only | 823 | 672 | 11.0 [5.1, 22.7] | 53.0% | 76.9% | 87.1% | 91.5% | 93.5% |
| Radiotherapy only | 1,000 | 825 | 8.0 [3.0, 17.7] | 64.6% | 82.3% | 88.6% | 92.2% | 94.0% |

Abbreviations: NSCLC: non-small cell lung cancer; CRT: Chemoradiotherapy; IQR: Interquartile range

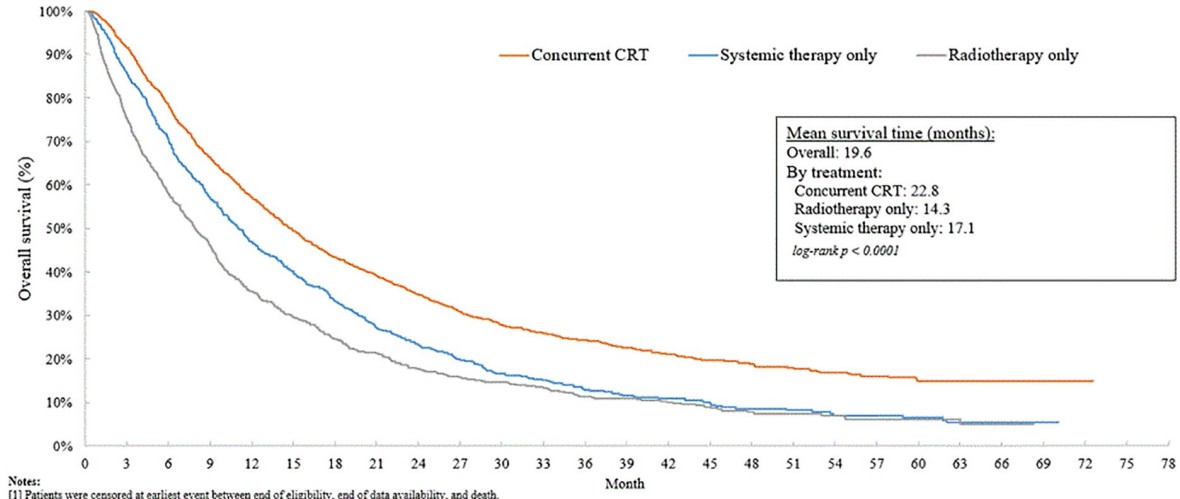

**Fig 2. Kaplan Meier curves for overall survival in unresected, stage III NSCLC patients treated with systemic therapy, radiotherapy, and concurrent chemoradiotherapy[1].**

65 years or older who received either cCRT or single modality therapy. Results of the current analysis indicate that only half of patients with unresected stage III NSCLC initiated treatment with cCRT. Generally, patients who initiated cCRT were younger, had a better predicted PS score, and a lower CCI compared to patients who received single modality therapy. This is in line with prior research conducted by Oh et al. (2017), which shows that cCRT is generally administered to healthier patients who are younger in age, and have minimal to no comorbidities, as well as the 2003 ASCO clinical practice guideline [17, 28].

In this analysis, patients receiving cCRT had significantly better survival compared to patients receiving single modality therapy, which may be in part due to younger age and better predicted PS. However, this analysis demonstrates that in an elderly population, patients with poor predicted PS and those with comorbidities also received cCRT, and overall survival was

**Table 4. Comparison of overall survival between unresected, stage III NSCLC patients treated with systemic therapy, chemotherapy, and concurrent chemoradiotherapy.**

| | Unadjusted Results | | Adjusted Results[1] | |
|---|---|---|---|---|
| | Hazard ratio (95% CI) | P-value | Hazard ratio (95% CI) | P-value |
| Systemic therapy only | 1.38 (1.26, 1.51) | <0.001* | 1.35 (1.22, 1.49) | <0.001* |
| Radiotherapy only | 1.75 (1.61, 1.91) | <0.001* | 1.58 (1.44, 1.73) | <0.001* |

Abbreviations: NSCLC: non-small cell lung cancer; CRT: Chemoradiotherapy; CI: Confidence interval

[1] Adjusted hazard ratios (HRs) were estimated from a Cox proportional hazards model adjusting for the following baseline covariates: race, gender, age, region, setting of residence, primary tumor site, grade, laterality, histology, stage (3a or 3b), index year of first therapeutic regimen, number of unique metastatic sites, all cancer metastatic sites at baseline, all comorbidities present at baseline, CCI, tumor marker tests, and predicted performance status.

improved in patients treated with cCRT versus single modality therapy even after adjusting for differences in these patient characteristics. Given the survival benefit with cCRT, older age and worse disease burden should not prohibit a patient from receiving cCRT if the physician deems them able to tolerate therapy. A 2019 meta-analysis conducted by Hung, et al. demonstrated better efficacy of CRT versus single-modality radiotherapy in patients with unresectable stage III NSCLC. However, the study included a younger patient population (54–77 years) while noting that data describing CRT outcomes in the elderly patient population is sparse and under-represented, and therefore, results of the study may not directly apply to the elderly patient population [29]. This study provides an important piece of real-world evidence to fill this gap in knowledge. Improved survival in elderly patients treated with cCRT versus single modality therapy is consistent with evidence-based clinical practice guidelines that designate cCRT (now, followed by durvalumab) as the current standard of care.

It was observed that elderly patients who received multidisciplinary care, specifically the 18% of patients who saw all three specialist types (medical oncologist, radiation oncologist, and surgical specialist) prior to the initiation of treatment, were more likely to receive cCRT over other therapy types. This is in line with the findings of Goulart, et al. (2013), which demonstrated that patients who saw physicians of multiple specialty types were most likely to receive cCRT in accordance with standard of care guidelines. Other studies have demonstrated improved quality and timeliness of care for NSCLC patients who receive multidisciplinary management, and who are referred to multiple types of cancer specialists [18, 27]. This study found that the proportion of elderly patients receiving cCRT over single modality therapy increased in the years after 2009, indicative of greater adoption of this standard of care in more recent years.

Receipt of cCRT may also allow the patient to receive novel therapies, such as durvalumab, which have recently become available to stage III NSCLC patients. Results from the PACIFIC trial (NCT02125461), a randomized, phase III, placebo-controlled trial, demonstrated that patients with stage III NSCLC who were treated with durvalumab following cCRT had significantly prolonged survival compared with placebo, with a hazard ratio of 0.68, with median overall survival not reached in the treatment group compared to 28.7 months in the placebo group, and a 24-month OS rate of 66.3% in the treatment group compared to 56.6% in the placebo group [30]. To date, there was limited research on the predictors of receiving treatment in the Medicare patients with unresectable, stage III NSCLC, and the current study provides an important piece of real-world evidence to fill this gap in knowledge.

Strengths of this study include the use of the SEER-Medicare database, which allows for the evaluation of tumor characteristics and treatment patterns of patients with unresected, stage III NSCLC in a real-world setting. The SEER-Medicare database includes a validated date of death which allows for a reliable and accurate evaluation of survival in the patient population. In addition, the use of the SEER cancer registries for diagnosis data diminishes misclassification bias from use of ICD-9-CM codes only, which is common in data sources only comprised of claims data. The combination of AMA Physician Masterfile allows physician characteristics to be linked to SEER-Medicare patient data, at the level of single claims. SEER-Medicare and the AMA Physician Masterfile are nationally representative databases, thus, results from this study are more generalizable as compared to healthcare claims database studies with data from specific states. This analysis went further than previous Medicare-based analyses of advanced NSCLC patients by assessing and including predicted PS in the primary analyses, an important confounder not otherwise available in claims data, as well as identified the use of therapy type using algorithms developed with expert clinical input.

The results of this study should be interpreted in the light of certain limitations. The SEER-Medicare database only includes data for those aged 65 years and older, and thus may not

capture all patients with NSCLC; therefore, study results are generalizable to those in the age-group eligible for Medicare coverage, and may not be applicable to other populations, such as younger patients enrolled in a commercial insurance plan. SEER-Medicare data are also released with a lag of several years, limiting the ability of this analysis to extend beyond 2014 at the time of the analysis; therefore, treatment patterns observed in this study may not reflect current trends. For example, patients included in this study were diagnosed with NSCLC between 2009 to 2013 and, during this time, it was not common to prescribe targeted therapy in stage III patients with EGFR or ALK mutations. Mutation-driven treatment emerged and became a more routine part of clinical practice only in recent years, with increased knowledge of disease.

Sites of metastases were identified using corresponding diagnosis codes from Medicare claims, as variables for sites of metastases in the SEER-Medicare data have not been fully vetted [31]. Note that this approach was developed by the study team for this project specifically in consultation with an oncologist, and was not one that was discussed, endorsed, nor validated by the SEER-Medicare team. Therefore, misclassification of patients with metastases and sites of metastases can occur based on miscoding of diagnoses, and these sites of metastases cannot be confirmed based on claims data alone; however, all patients with more than one primary tumor (i.e., primary tumors other than lung cancer) were excluded, thus any codes for malignancies other than lung cancer were assumed to be indicative of metastases to other sites. Given the limitations of these data and the fact that this assumption could not be definitively validated, results on sites of metastases should be interpreted with caution. In the analysis of multidisciplinary care, not all specialists who were involved in a patient's care may be reflected in claims data, for example, if a patient's case and treatment plan was discussed at a multidisciplinary tumor board. Additionally, claims for RT do not contain the level of granularity required to distinguish between treatments administered with curative intent compared to those administered with palliative intent. As with any observational study, because of its non-randomized nature, this study may be subject to residual confounding due to unmeasured confounders.

## Conclusion

cCRT was the standard of care for the treatment of unresectable stage III NSCLC prior to the PACIFIC trial (which added subsequent treatment with durvalumab to standard of care). This retrospective analysis shows that cCRT was the initial treatment in approximately 50% of patients during the study period of 2009–2014. A number of demographic and clinical factors contributed to treatment selection, including age and health status of the patient, and whether the patient received multidisciplinary care. Patients who received multidisciplinary care were more likely to receive cCRT over single modality therapy. However, less than a quarter of even the cCRT group saw all three physician specialties before starting treatment, indicating that there is room for improvement in delivering multidisciplinary patient care. Given the survival benefit of receiving cCRT over single-modality therapy, physicians should be encouraged to pursue cCRT in all appropriate patients with unresectable stage III NSCLC.

## Acknowledgments

This study used the linked SEER-Medicare database. The interpretation and reporting of these data are the sole responsibility of the authors. The authors acknowledge the efforts of the National Cancer Institute; the Office of Research, Development and Information, CMS; Information Management Services (IMS), Inc.; and the Surveillance, Epidemiology, and End Results (SEER) Program tumor registries in the creation of the SEER-Medicare database.

## Author Contributions

**Conceptualization:** Priyanka Bobbili, Kellie Ryan, Maral DerSarkissian, Mei Sheng Duh, Jorge E. Gomez.

**Data curation:** Priyanka Bobbili, Maral DerSarkissian, Akanksha Dua, Christopher Yee.

**Formal analysis:** Priyanka Bobbili, Akanksha Dua, Christopher Yee.

**Funding acquisition:** Kellie Ryan, Mei Sheng Duh.

**Investigation:** Kellie Ryan, Maral DerSarkissian.

**Methodology:** Priyanka Bobbili, Kellie Ryan, Maral DerSarkissian, Akanksha Dua, Christopher Yee, Jorge E. Gomez.

**Project administration:** Priyanka Bobbili, Kellie Ryan, Maral DerSarkissian, Mei Sheng Duh, Jorge E. Gomez.

**Resources:** Priyanka Bobbili, Kellie Ryan, Maral DerSarkissian, Mei Sheng Duh.

**Software:** Christopher Yee.

**Supervision:** Priyanka Bobbili, Kellie Ryan, Maral DerSarkissian, Jorge E. Gomez.

**Validation:** Priyanka Bobbili, Maral DerSarkissian, Akanksha Dua, Christopher Yee.

**Visualization:** Priyanka Bobbili, Kellie Ryan, Maral DerSarkissian, Akanksha Dua, Christopher Yee, Mei Sheng Duh, Jorge E. Gomez.

**Writing – original draft:** Priyanka Bobbili, Maral DerSarkissian, Akanksha Dua, Christopher Yee.

**Writing – review & editing:** Priyanka Bobbili, Kellie Ryan, Maral DerSarkissian, Akanksha Dua, Christopher Yee, Mei Sheng Duh, Jorge E. Gomez.

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
