## [Decision Letter · Decision Letter 0]

13 Jan 2020

PONE-D-19-31531

Predictors of chemoradiotherapy versus single modality therapy and overall survival among patients with unresectable, stage III non-small cell lung cancer

PLOS ONE

Dear Dr. Bobbili,

Thank you for submitting your manuscript to PLOS ONE. After careful consideration, we feel that it has merit but does not fully meet PLOS ONE’s publication criteria as it currently stands. Therefore, we invite you to submit a revised version of the manuscript that addresses the points raised during the review process.

ACADEMIC EDITOR: The present study is important in this field of expertise. I sincerely request the authors to respond to the valuable questions raised from our reviewers.  

We would appreciate receiving your revised manuscript by Feb 27 2020 11:59PM. To enhance the reproducibility of your results, we recommend that if applicable you deposit your laboratory protocols in protocols.io, where a protocol can be assigned its own identifier (DOI) such that it can be cited independently in the future. For instructions see: http://journals.plos.org/plosone/s/submission-guidelines#loc-laboratory-protocols

We look forward to receiving your revised manuscript.

Kind regards,

Jason Chia-Hsun Hsieh, M.D. Ph.D

Academic Editor

PLOS ONE

Additional Editor Comments (if provided):

Please respond to the questions raised by the reviewers.

Journal Requirements:

2. We noticed you have some minor occurrence(s) of overlapping text with the following previous publication(s), which needs to be addressed:

https://doi.org/10.2217/fon-2019-0282

https://doi.org/10.1002/pds.4864

In your revision ensure you cite all your sources (including your own works), and quote or rephrase any duplicated text outside the Methods section. Further consideration is dependent on these concerns being addressed.

"This research was funded by AstraZeneca, Gaithersburg, MD, USA."       

We note that one or more of the authors have an affiliation to the commercial funders of this research study : 'AstraZeneca'

Reviewers' comments:

Reviewer's Responses to Questions

5. Review Comments to the Author

Reviewer #1: This manuscript would like to find out the predictors of cCRT and survival outcomes in stage III NSCLC and showed the patients undergoing cCRT were younger and had good PS and better survival outcomes than in patients undergoing RT or ST only. This study is well-written and reflects the real-world experiences. However, I have some comments that authors might have to clarify these points.

1. As authors mentioned that SEER-Medicare data only included the patients aged ≥65 years, the authors studied in the population of the elderly. Therefore, all the findings in this manuscript should be limited in the elderly rather than whole population.

2. In the methods, authors stated the systemic therapy (ST) included chemotherapy and targeted. As we know that the patients undergoing targeted therapy or chemotherapy as initial treatment are completely different sub-population (with or without driver gene mutation) with distinct survival outcomes. Therefore, the authors should separate the two groups in all the analysis or exclude targeted therapy group.

3. In figure 1, the authors exclude the patients undergoing surgical resection within 4 months after diagnosis and did not explain the reason in the manuscript. In clinical practice, some patients experience very good response after initial treatment so these patients could receive surgical resection. Therefore, could authors explain the reason why these patients were excluded.

4. The authors stated “Prior to the initiation of treatment, medical oncologists were most frequently seen (78.1%) followed by surgical specialists (47.8%) then radiation oncologists (42.8%).” However, nearly 78% of patients received radiotherapy therapy in current study (including cCRT and RT groups). I suppose that radiotherapy should be prescribed by radiation oncologists. Could authors explain why only 42.8% visited radiation oncologists in current cohort?

5. In the table 1, I don’t really know what “Number of unique metastatic sites” means? In term of stage III NSCLC, the patients should not have metastatic sites?

Reviewer #2:

1. All stages were suggested to present, according to the 8th ed AJCC TNM staging system. The definition of stage III was different from the 6th and 7th edition version.

2. Stage IIIa and stage IIIB lung cancers have different disease severity. Staging migration existed among different staging systems. Besides, the treatment among inoperable stages 3a and 3b were different. Chemotherapy with or without radiotherapy was mainly used in stage IIIa, and targeted therapy was used primarily in stage IIIb patients with EGFR or ALK mutation(s). The difference should be addressed in the manuscript.

3. Patients who presented with T3 satellite N1-3M0 and T4 ipsilateral nodule N1-3M0 were all excluded, and any accounted up to one-third of the stage III population. The exclusion may weaken representative of the study cohort and result in a biased conclusion. The selection bias has to be minimized and addressed in the manuscript.

Reviewer #3: The authors should be commended for their efforts to collect data from a large cohort of patients with NSCLC. The manuscript is well written and the topic is relevant. However, as they indicate in the discussion, the retrospective nature of this analysis limits the effectiveness of the conclusion. Moreover, the manuscript does not add anything new to the literature. The most interesting data are related to the fact that half of the patients evaluated received concurrent chemotherapy, despite we do not know the dose of radiotherapy given to the patients and the number of chemotherapy cycles administered concurrently to radiation, or the chemotherapy regimens used. Another interesting point is that only a minority of the patients (17%) was seen by a multidisciplinary team before starting treatment, and among these a larger proportion of cases received concurrent chemo-radiotherapy, suggesting the importance of multidisciplinary discussion before choosing treatment.

I recommend to better highlight what emerges from this retrospective analysis in comparison to previous published papers because it is not clear.

6. PLOS authors have the option to publish the peer review history of their article (what does this mean?). If published, this will include your full peer review and any attached files.

Reviewer #1: Yes: Chiao-En Wu

Reviewer #2: No

Reviewer #3: No

---

## [Author Response · Author response to Decision Letter 0]

21 Feb 2020

Reviewer#1

Comments to the Author

1. As authors mentioned that SEER-Medicare data only included the patients aged ≥65 years, the authors studied in the population of the elderly. Therefore, all the findings in this manuscript should be limited in the elderly rather than whole population.

Response: 

Thank you for sharing this comment; we agree that the results of our study are most relevant to an elderly patient population. We would like to bring the reviewers’ attention to the limitations section of our manuscript where we state the following: 

“The results of this study should be interpreted in the light of certain limitations. The SEER-Medicare database only includes data for those aged 65 years and older, and thus may not capture all patients with NSCLC; therefore, study results are generalizable to those in the age-group eligible for Medicare coverage, and may not be applicable to other populations, such as younger patients enrolled in a commercial insurance plan.”

In addition, we have made modifications to the discussion section of the manuscript to emphasize that the findings of the manuscript are limited to an elderly population (≥65 years old). These are noted as tracked changes in the revised manuscript. 

2. In the methods, authors stated the systemic therapy (ST) included chemotherapy and targeted. As we know that the patients undergoing targeted therapy or chemotherapy as initial treatment are completely different sub-population (with or without driver gene mutation) with distinct survival outcomes. Therefore, the authors should separate the two groups in all the analysis or exclude targeted therapy group.

Response:

Thank you for this insightful comment. This study included patients with stage III NSCLC who were diagnosed during the period 2009-2013. During this time, while mutational testing was part of standard clinical practice for patients with metastatic disease, it was not standard clinical practice for patients with locally advanced disease who were candidates for chemoradiation. Even now testing is more applicable to metastatic disease. Thus, the mutational status of these patients is unlikely to have been known, and unlikely to have been a driver for receipt of targeted therapy. 

Additionally, the main goal of this study was to investigate predictors of receiving concurrent chemoradiotherapy (CRT) compared to single-modality therapy. For example, in Table 3 (Logistic regression to identify predictors of CRT vs. single-modality therapy among patients with unresected, stage III NSCLC patients), all patients who received single-modality therapy i.e. systemic therapy (ST; comprised of chemotherapy-only and targeted therapy-only patients), and radiotherapy (RT) are grouped together, and compared to the cohort of patients receiving CRT. The results of this analysis indicate that compared to patients that receive single-modality therapy (whether it be chemotherapy, targeted therapy, or RT), patients who receive CRT are more likely to be younger, healthier, and have better performance status. The inclusion of patients who received targeted therapy in the single modality group could potentially improve the survival of that group if these patients had a driver mutation. Thus, including patients who received targeted therapy (as a surrogate for having a driver mutation) in the single-modality group, and still finding a survival benefit in the CRT group strengthens the findings of this analysis. 

Lastly, we conducted a sensitivity analysis for the comparison of overall survival between patients treated with ST, RT, and CRT (Table 4 in the manuscript) after excluding patients treated with targeted therapy. Please find the comparison of the results tables in the uploaded Word document containing response to reviewer comments. 

The unadjusted and adjusted results after excluding targeted therapy patients are almost identical to the original results in the manuscript, further demonstrating that comparison of overall survival between the ST and CRT treatment groups is not impacted by the presence of targeted therapy patients in the ST group (i.e., due to the disease stage and time period of diagnosis of the study population). This sensitivity analysis demonstrates the robustness of the results presented in the manuscript, and as such we would recommend to keep the current analyses unchanged in the manuscript. 

3. In figure 1, the authors exclude the patients undergoing surgical resection within 4 months after diagnosis and did not explain the reason in the manuscript. In clinical practice, some patients experience very good response after initial treatment so these patients could receive surgical resection. Therefore, could authors explain the reason why these patients were excluded.

Response: 

Thank you for this comment. The goal of our research was to study patients that had locally advanced lung cancer and for whom surgery was not a treatment option. Therefore, surgical resection of tumor was defined as an exclusion criterion in our study, as noted in the Materials and Methods section of the manuscript:

“Patients were excluded from the analysis if the reporting source of their lung cancer diagnosis was an autopsy or death certificate, if they had more than one primary tumor, or if they had surgical resection of their tumor. Surgical resection was identified using both surgical history captured in the SEER registries, which noted whether a cancer-directed surgery was performed within 4 months of NSCLC diagnosis date, and surgical procedures identified using Medicare claims”

Receipt of cancer-directed surgery within 4 months of NSCLC diagnosis was the standard definition of surgical treatment in SEER data. 

4. The authors stated “Prior to the initiation of treatment, medical oncologists were most frequently seen (78.1%) followed by surgical specialists (47.8%) then radiation oncologists (42.8%).” However, nearly 78% of patients received radiotherapy therapy in current study (including cCRT and RT groups). I suppose that radiotherapy should be prescribed by radiation oncologists. Could authors explain why only 42.8% visited radiation oncologists in current cohort?

Response: 

Thank you for this thoughtful question. This analysis evaluates visits to specialists prior to the initiation of anti-cancer treatment. Therefore, it is possible that more patients visited radiation oncologists after the initiation of their treatment, and thus were prescribed treatment with radiation therapy at that time. Our findings show that patients who saw all three specialist types prior to the initiation of treatment were more likely receive treatment with CRT, which is the recommended standard-of-care treatment for this patient population. 

5. In the table 1, I don’t really know what “Number of unique metastatic sites” means? In term of stage III NSCLC, the patients should not have metastatic sites?

Response: 

Thank you for your question. It is possible for patients with stage III NSCLC to have metastatic sites in the lung and lymph nodes, which was observed in over half of the patients in our dataset (‘others’ categories included <10% of patients, with the exception of patients with unspecified metastatic sites [~15%]). 

This is in line with the TNM classification, which indicates that stage III NSCLC can include cancers that have spread to lymph nodes near the collar bone, around the carina, the mediastinum, the heart, large blood vessels near the heart, the diaphragm, backbone, or trachea. 

To address this comment, we have added clarification to footnote 9 of Table 1 (Comparison of baseline characteristics of unresected, stage III NSCLC patients treated with systemic therapy, radiotherapy, and concurrent chemoradiotherapy) in the tracked change manuscript:

“Metastatic sites were evaluated based on ICD-9 codes. Diagnoses of other malignancies were considered to be metastases since all patients in the study population were required to only have 1 primary tumor (NSCLC). Metastatic sites included but were not limited to diagnosis codes for lymphatic and hematopoietic tissue cancer, respiratory cancer (excluding cancer of the lungs), bone and bone marrow cancer, brain and spinal cord cancer, tissue cancer, and endocrine cancer among others.”

Reviewer#2

1. All stages were suggested to present, according to the 8th ed AJCC TNM staging system. The definition of stage III was different from the 6th and 7th edition version.

Response: 

Thank you for your comment. The objective of this study was to understand the predictors of treatment (single-modality versus concurrent chemoradiotherapy) and overall survival among patients with unresectable stage III non-small cell lung cancer. As outlined in our introduction, stage III NSCLC patients comprise almost one-third of all NSCLC patients, and present with low survival rates (13%-36%). Despite progress in the treatment landscape and current recommendations for treatment with multimodality therapy, patients with stage III NSCLC are often treated with single modality chemotherapy or radiotherapy instead. It was the goal of our study to understand why the stage III patient population does not receive multimodality therapy (such as chemoradiotherapy). 

Patients diagnosed with NSCLC between January 1, 2009 to December 31st, 2013 were included in this study. We used different versions of the AJCC TNM staging system to correspond to the staging criteria available at time of diagnosis for the patients in our study. Thus, the version in use at the time of diagnosis (either AJCC TNM 6th edition or AJCC TNM 7th edition) would have driven the classification of staging, (and also the treatment received by the patient). Therefore, patients were classified according to the 6th edition if diagnosed in 2009) or the 7th edition if diagnosed between 2010 and 2013. This is noted in the “Study population” section of our manuscript, as state listed below: 

“Patients were further required to have stage III disease at the time of diagnosis, based on the American Joint Committee on Cancer (AJCC) Tumor, Nodes, Metastases (TNM) staging system, 6th edition (for patients diagnosed in 2009) or 7th edition (for patients diagnosed in 2010 - 2013).” This was done to ensure we identified all potential patients who would have been identified and considered for CRT as part of their treatment regimen. Additionally, all patients diagnosed with stage III NSCLC according to the AJCC 6th edition would be classified as stage III NSCLC according to the AJCC 7th edition. This indicates that there was likely no misclassification of stage III patients in the current analysis, and ensures that the use of two classification systems did not impact the observed results. 

2. Stage IIIa and stage IIIB lung cancers have different disease severity. Staging migration existed among different staging systems. Besides, the treatment among inoperable stages 3a and 3b were different. Chemotherapy with or without radiotherapy was mainly used in stage IIIa, and targeted therapy was used primarily in stage IIIb patients with EGFR or ALK mutation(s). The difference should be addressed in the manuscript.

Response: 

Thank you for your thoughtful comment. Patients included in this study were diagnosed with NSCLC between 2009 to 2013. During this time, it was not common to prescribe targeted therapy in stage IIIB patients with EGFR or ALK mutations. This is largely because mutation-driven treatment emerged, and became a more routine part of clinical practice, later in the decade. We have added wording to note this in the limitations section of our manuscript. 

Additionally, we would like to highlight that in our current analyses, within the stage IIIA subgroup of patients (N = 2,260), approximately 54% received CRT, 19% received ST, and 28% RT, whereas among stage IIIB patients (N = 1,539), approximately 50% of patients received CRT, 26% of patients received ST, and 24% received RT. This indicates that the proportion of patients receiving CRT in the stage IIIA subgroup (54%) is comparable to the proportion of patients receiving chemoradiotherapy in the stage IIIB subgroup (50%). Given this finding, it is unlikely that the differences in the observed clinical outcomes are solely due to the distribution of stage IIIA and IIIB patients in the cohorts. 

We would also like to note that stage (IIIA or IIIB) was included as one of the variables for adjustment in the multivariable cox model, and the adjustment procedure would have potentially accounted for any imbalances in patient characteristics/underlying staging. This would indicate that the estimated differences between the cohorts are likely due to the type of treatment and care received. 

3. Patients who presented with T3 satellite N1-3M0 and T4 ipsilateral nodule N1-3M0 were all excluded, and any accounted up to one-third of the stage III population. The exclusion may weaken representative of the study cohort and result in a biased conclusion. The selection bias has to be minimized and addressed in the manuscript

Response: 

Thank you for this comment. The inclusion criteria for this study required patients to have stage III cancer at time of diagnosis, based on the AJCC TNM staging system available at time of diagnosis. Exclusion of patients who presented with T3 satellite N1-3MO and T4 ipsilateral nodule N1-3M0 was not a specific criterion for the study. Rather, these patients were eligible for inclusion in the study population if they did not have evidence of surgical resection. 

Patients with a surgical resection of tumor were excluded from the study, and they comprised ~6% of the stage III sample population. 

Reviewer#3

1. The authors should be commended for their efforts to collect data from a large cohort of patients with NSCLC. The manuscript is well written and the topic is relevant. However, as they indicate in the discussion, the retrospective nature of this analysis limits the effectiveness of the conclusion. Moreover, the manuscript does not add anything new to the literature. The most interesting data are related to the fact that half of the patients evaluated received concurrent chemotherapy; despite we do not know the dose of radiotherapy given to the patients and the number of chemotherapy cycles administered concurrently to radiation, or the chemotherapy regimens used. Another interesting point is that only a minority of the patients (17%) was seen by a multidisciplinary team before starting treatment, and among these a larger proportion of cases received concurrent chemo-radiotherapy, suggesting the importance of multidisciplinary discussion before choosing treatment. I recommend to better highlight what emerges from this retrospective analysis in comparison to previous published papers because it is not clear.

Response: 

Thank you for the feedback and the insightful comments. We agree with your point to highlight the conclusions of the current study in the context of previous research. We have added languages for the same to the discussion section of the manuscript, in tracked changes: 

“A 2019 meta-analysis conducted by Hung, et al. demonstrated better efficacy of CRT versus single-modality RT in patients with unresectable stage III NSCLC. However, the study included a younger patient population (54-77 years) while noting that data describing CRT outcomes in the elderly patient population is sparse and under-represented, and therefore, results of the study may not directly apply to the elderly patient population [18]. The current study provides an important piece of real-world evidence to fill this gap in knowledge.”

Additionally, please find language in the discussion and conclusion to further highlight the importance of receiving multidisciplinary treatment in the context of prior research.

“It was observed that elderly patients who received multidisciplinary care, specifically the 18% of patients who saw all three specialist types (medical oncologist, radiation oncologist, and surgical specialist) prior to the initiation of treatment, were more likely to receive cCRT over other therapy types. This is in line with the findings of Goulart, et al. (2013), which demonstrated that patients who saw physicians of multiple specialty types were most likely to receive cCRT in accordance with standard of care guidelines. Other studies have demonstrated improved quality and timeliness of care for NSCLC patients who receive multidisciplinary management, and who are referred to multiple types of cancer specialists [17, 25].”

“Patients who received multidisciplinary care were more likely to receive cCRT over single modality therapy. However, less than a quarter of even the cCRT group saw all three physician specialties before starting treatment, indicating that there is room for improvement in delivering multidisciplinary patient care.”

---

## [Editor Report · Decision Letter 1]

2 Mar 2020

Predictors of chemoradiotherapy versus single modality therapy and overall survival among patients with unresectable, stage III non-small cell lung cancer

PONE-D-19-31531R1

Dear Dr. Bobbili,

We are pleased to inform you that your manuscript has been judged scientifically suitable for publication and will be formally accepted for publication once it complies with all outstanding technical requirements.

With kind regards,

Jason Chia-Hsun Hsieh, M.D. Ph.D

Academic Editor

PLOS ONE

Additional Editor Comments (optional):

All the questions were addressed adequately.
---

## [Editor Report · Acceptance letter]

5 Mar 2020

PONE-D-19-31531R1 

Predictors of chemoradiotherapy versus single modality therapy and overall survival among patients with unresectable, stage III non-small cell lung cancer 

Dear Dr. Bobbili:

I am pleased to inform you that your manuscript has been deemed suitable for publication in PLOS ONE. Congratulations! Your manuscript is now with our production department. 

With kind regards,

on behalf of

Dr. Jason Chia-Hsun Hsieh 

Academic Editor

PLOS ONE